# Research on Road Internal Disease Identification Algorithm Based on Attention Fusion Mechanisms

**DOI:** 10.3390/s24206757

**Published:** 2024-10-21

**Authors:** Yangyang Wang, Shoujing Yan, Chenchen Xi, Zhi Yu, Chunpeng Zhou, Fengxia Chi, Jintao Wei

**Affiliations:** 1Department of Architecture and Civil Engineering, Zhejiang University, Hangzhou 310030, China; 12012139@zju.edu.cn; 2Institute of Road Engineering, Zhejiang Scientific Research Institute of Transport, Hangzhou 310023, China; zjjk@zjjtkyy.com (C.X.); wujj0329@163.com (F.C.); weijintao2020@163.com (J.W.); 3Zhejiang Province Key Laboratory of Road and Bridge Detection and Maintenance Technology Research, Zhejiang Scientific Research Institute of Transport, Hangzhou 310023, China; 4Zhejiang Provincial Key Laboratory of Service Robot, School of Software Technology, Zhejiang University, Hangzhou 310027, China; yuzhirenzhe@zju.edu.cn (Z.Y.); zhoucp@zju.edu.cn (C.Z.)

**Keywords:** road health monitoring, deep learning, multi-view identification, attention fusion mechanism

## Abstract

Internal disease in asphalt pavement is a crucial indicator of pavement health and serves as a vital basis for maintenance and rehabilitation decisions. It is closely related to the optimization and allocation of funds by highway maintenance management departments. Accurate and rapid identification of internal pavement diseases is essential for improving overall pavement quality. This study aimed to identify internal pavement diseases using deep learning algorithms, thereby improving the efficiency of determining internal pavement diseases. In this work, a multi-view recognition algorithm model based on deep learning is proposed, with attention fusion mechanisms embedded both between channels and between views. By comparing and analyzing the training and recognition results of different neural networks, it was found that the multi-view recognition algorithm model based on attention fusion demonstrates the best performance in identifying internal pavement diseases.

## 1. Introduction

Asphalt pavement is subjected to high-frequency traffic loads and is exposed to harsh natural environments for extended periods, leading to various forms of damage on the surface and within [1,2,3,4], such as cracks, potholes, and subsidence. Damage to asphalt pavements reduces driving quality and comfort, reduced driving safety, and raises the cost of pavement maintenance [5,6]. Therefore, to extend the service life of highways and improve the quality of highway services, the rapid detection and identification of internal pavement diseases is an important task for highway maintenance management departments.

Internal diseases in asphalt pavements are detected using ground-penetrating radar (GPR 3.6.0) [7,8], where images of internal pavement diseases are generated via radar electromagnetic waves. On the radar map, internal pavement diseases exhibit features such as high brightness, hyperbolas, and dark spots in the front view (B-Scan) and top view (C-Scan) [9,10,11]. For internal pavement diseases, manual identification is usually used for judgment. However, the manual identification method is based on subjective experience, which has a significant impact on the accuracy of identifying internal pavement diseases.

In recent years, with the development of artificial intelligence algorithms, some researchers have proposed new algorithms for the identification of internal road diseases [12,13,14]. Targeting internal road voids, pipelines, cracks, and other diseases, Kang et al. integrated the three-view features of 3D ground-penetrating radar and proposed a neural network recognition model based on 3D views [15]. Additionally, by developing a background filtering algorithm based on basis pursuit, they filtered the radar data, resolving the interference of underground clutter signals on radar imaging. Li et al. [16] proposed a deep learning method for underground road diseases and conducted model training and verification on more than 270 km of road. The proposed model obtained an accuracy rate of 72.39% and 68.74% for the identification of crack diseases and void diseases inside roads, respectively. Tong et al. [17] used multiple cascaded neural network models to detect and identify internal road diseases, such as cracks, water erosion pits, and uneven settlements, achieving an accuracy rate of over 85%. However, the recognition accuracy and efficiency of internal road disease identification algorithms are relatively low, making it difficult for them to meet the requirements for the rapid detection of internal road conditions.

To improve the accuracy and recognition efficiency of internal road diseases, Liu et al. combined the you only live once (YOLO) series with 3D ground-penetrating radar images to establish an internal road disease identification model based on YOLOv5 [18]. A benefit evaluation of maintenance programs based on these two detection methods was conducted from economic and environmental perspectives. The results demonstrated that the economic scores improved and that the maintenance cost was reduced by USD 49,398/km based on GPR detection; the energy consumption and carbon emissions were reduced by 792,106 MJ/km (16.94%) and 56,289 kg/km (16.91%), respectively. Gao et al. [19] proposed Faster R-Convent, achieving 89.13% accuracy and 86.24% Itou for radar images of internal road diseases. In view of the hyperbolic features on radar maps, Zhang et al. [20] proposed a deep learning framework based on generative adversarial networks and compared it with region-based convolutional neural networks (RCNN), Cascade R-CNN, the single-shot multibox detector (SSD), and YOLOv2 neural networks. The proposed method was found to be superior to existing methods, with an average accuracy rate of 97%. Hou et al. [21] proposed a deep neural network (DNN) architecture called GPRInvNet, which solves the problem of spatial alignment between time series scan data and spatial dielectric constant maps, and it can effectively reconstruct internal road diseases with clear boundaries. Li et al. [22,23] proposed an effective method for the automatic identification and positioning of hidden cracks based on 3D ground-penetrating radar and deep learning models, and they used YOLOv3, YOLOv4, and YOLOv5 for identification. The best mean average precision (mAP) of the YOLOv5 model reached 94.39%, and the YOLOv4 model showed better robustness than the YOLOv5 model and could accurately distinguish between hidden cracks and pseudo-cracks. However, these methods cannot comprehensively extract the features of internal diseases in pavement. In recent years, extensive reports have demonstrated that attention mechanisms play a critical role in algorithms for image recognition, natural language processing, and other domains [24,25,26]. By integrating attention mechanisms, models gain the ability to selectively concentrate on the most relevant components of the input data, rather than assigning uniform importance. This is particularly noticeable in object detection tasks, where attention mechanisms effectively prioritize key features by assigning differential weights to different regions. This targeted approach not only streamlines computational processes but also boosts the precision of the model by facilitating the integration of multi-view features. Therefore, in this study, we proposed a multi-view algorithm model that incorporates attention fusion mechanisms between channel-wise and between element-wise, designed to enhance the efficiency and accuracy of the algorithm model in recognizing internal pavement diseases.

In this paper, we propose an end-to-end deep attention twin-tower model that utilizes both views of B-Scan and C-Scan, and its recognition results are compared with those of traditional recognition algorithms. In Section 2, the framework of the multi-view recognition algorithm model, as well as the attention fusion mechanisms between channel-wise and between element-wise, is introduced. In Section 3, the experimental process and results of the algorithm model’s recognition are detailed, including the collection of experimental data, the setting of experimental parameters, the testing results of various algorithm models on the original dataset and the augmented dataset, and the generalization ability of the algorithm. Section 4 summarizes the research.

## 2. Methodology

Internal pavement diseases are examined using ground-penetrating radar, and the radar images include two dimensions: B-Scan and C-Scan. Numerous research findings indicate that training recognition algorithms based on a single view fail to fully extract the features of internal pavement diseases, which reduces the accuracy of disease identification [27,28]. This study used B-Scan and C-Scan imaging data obtained from ground-penetrating radar as input, training feature extractors on each view separately. Then, the features from both views were fused, followed by the classification of internal pavement diseases, as shown in Figure 1. When extracting features from a single view, an inter-channel self-attention mechanism was used to learn the weight distribution between channels, thereby performing weighted addition for feature fusion.

To better extract disease feature information from each view while suppressing the expression of useless information such as image noise in the model, this study introduced an inter-channel attention mechanism in the transition layers between each block of the dense convolutional network (DenseNet) and the final classification layer to learn the weight distribution between channels, as shown in Table 1. Subsequently, the attention mechanism was combined again to achieve the weighted fusion of dual-view features. Finally, in the classification layer of the dual-tower model, dual-view feature maps were combined for road disease classification. The specific application of these two attention mechanisms to radar map data is introduced in Section 2.2 and Section 2.3.

This study utilized the DenseNet-121 network [29] as the basic architecture to extract information from three views. Each view was trained with a separate feature extractor, and these extractors do not share parameters. The network structure of the single-view feature extractor is shown in Table 1, mainly comprising four dense blocks that maximize feature utilization, transition layers that reduce dimensionality, and a representation layer that fuses dual-view feature maps. The dense blocks include multiple convolution operations and introduce 1 × 1 convolution operations to improve computational efficiency. The transition layers use convolution, inter-channel self-attention fusion mechanisms, pooling, and other operations to reduce the number of channels and decrease the image size. The representation layer consists of inter-channel self-attention fusion mechanisms, global average pooling, dual-view feature fusion, fully connected layers, SoftMax, and other operations, ultimately outputting the road disease prediction results.

### 2.1. Channel-Wise Attention

To reduce the focus of the feature extractor on irrelevant information and to make it pay more attention to the disease feature information in radar map data, this study adopted a channel-wise attention mechanism [30,31] to learn the weight size between different channels, and it used the learned channel feature importance distribution to calibrate the original features, reflecting the importance of different channels.

If a given view is used, by taking the intermediate feature map $U\in R^{C\times H\times W}$ as input, the entire inter-channel self-attention process can be represented by Equation (1):(1)$$\begin{array}{c} \bar{U}=M_{c}⊗U \end{array}$$

Here, $\bar{U}\in R^{C\times H\times W}$ represents the output of the attention process, $\;M_{c}\in R^{C\times 1\times 1}$ denotes the one-dimensional channel attention distribution that needs to be learned, and ⊗ refers to element-wise multiplication. To enable the view feature extractor to automatically learn the importance relationships between channels in the feature maps, this study employed a squeeze-and-excitation (SE) structure based on Inception to implement the inter-channel attention mechanism. The SE module mainly consists of two operations: squeeze and excitation. It enhances the network’s expressive capability by obtaining the distribution of importance between feature channels, and its specific structure is shown in Figure 2.

The SE module performs a squeeze operation on the input feature map, using global average pooling to encode all spatial features in a single channel into a global feature vector $v\in R^{C}$ based on channel relationships. Given an input feature map U with spatial dimensions of C×H×W—where C represents the number of channels, and H and W represent the height and width, respectively—we denote the c-th feature map in U as $U_{c}$. After the squeeze operation, the inter-channel global feature vector v can be represented by Formula (2):(2)$$\begin{array}{c} v_{c}=\frac{1}{H\times W}\overset{H}{\underset{i=1}{\sum }}\overset{W}{\underset{j=1}{\sum }}u_{c}\left(i,j\right) \end{array}$$

In the SE module, the excitation operation employs a gating mechanism to adaptively learn the nonlinear relationships between different channels. Additionally, the bottleneck structure, which consists of two fully connected layers, not only constrains the complexity of this operation but also helps to enhance the model’s generalization capability. The global feature v is input into a fully connected layer with weights $W_{1}$, followed by activation using the nonlinear RELU function, where RELU(x)=max(x,0). Subsequently, a fully connected layer with weights $W_{2}$ is utilized once more to restore the initial dimensions, and, finally, the output is produced through the nonlinear Sigmoid function. The Sigmoid function can be represented as σ(x)=1/(1+exp(−x)). The feature attention distribution *M* between channels can be represented by Formula (3), given the dimensionality reduction coefficient r, $W_{1}\in R^{\frac{C}{r}\times C}$, and $W_{2}\in R^{C\times \frac{C}{r}}$:(3)$$\begin{array}{c} M=\sigma \left(W_{2}\;RELU\left(W_{1}v\right)\right) \end{array}$$

Finally, the learned feature attention distribution M is used to rescale the original input features U. The feature map and feature attention distribution corresponding to the c-th element are denoted as $m_{c}$ and $u_{c}$, respectively. The c-th element ${\bar{u}}_{c}$ of the output result can be represented by Equation (4):(4)$$\begin{array}{c} {\bar{u}}_{c}=m_{c}\cdot u_{c} \end{array}$$

Due to its simple structure, fewer parameter settings, and lower computational complexity, the SE module can be added to different network architectures. Therefore, this study incorporates the SE module into the transition and classification layers of DenseNet, as shown in Figure 3, to enhance the network’s ability to express road distress information and simultaneously suppress irrelevant features.

### 2.2. Element-Wised Attention

In Figure 1, it can be observed that different types of diseases have various degrees of emphasis on the information presented in the B-Scan and C-Scan views. For example, cavity diseases primarily rely on the highlighted disease information in B-Scan for judgment, whereas crack problems require the dual representation of hyperbolas in B-Scan and elongated shapes in C-Scan for an accurate assessment. Therefore, an inter-view attention mechanism, specifically element-wise attention [32,33], is employed to learn the importance of different views during the feature fusion process to assign different fusion weights to various samples. This enhances the weight of significant views and reduces that of less important ones, resulting in a set of weight values for the inter-view fusion of each sample. Subsequently, the weighted sum of C-Scan and B-Scan is calculated using the determined weight values, achieving an adaptive weighted fusion of the features from both views. The process of dual-view feature fusion is illustrated in Figure 4.

If the feature maps extracted from C-Scan and B-Scan are denoted as $A\in R^{N\times d}$ and $B\in R^{N\times d}$, respectively, and N represents the batch size, the complete process of view fusion using weighted addition can be expressed by Equation (5):(5)$$\begin{array}{c} F=\alpha A+\beta B \end{array}$$

Here, $F\in R^{N\times d}$ represents the features after merging the two views, and the attention weight values $\alpha ,\beta \in R^{N}$ between the views are determined by the view attention mechanism. The specific calculation process is shown in Formulas (6)–(9):(6)$$\begin{array}{l} \begin{array}{c} \bar{\alpha }=q_{att}\tanh\left(W_{att}A+b_{att}\right) \end{array} \end{array}$$
(7)$$\begin{array}{c} \bar{\beta }=q_{att}\tanh\left(W_{att}B+b_{att}\right) \end{array}$$
(8)$$\begin{array}{c} \alpha ⊕\beta =SoftMax\left(\bar{\alpha }⊕\bar{\beta }\right) \end{array}$$
(9)$$\begin{array}{c} SoftMax\left(x\right)=\frac{\exp\left(x\right)}{\sum_{c=1}^{C}\exp\left(x_{c}\right)} \end{array}$$

Here, $\bar{\alpha },\bar{\beta }\in R^{N}$ are calculated using the attention mechanism of the C-Scan and B-Scan feature maps, respectively, including two linear transformations and the hyperbolic tangent function (tanh). For different samples, intermediate results $\bar{\alpha },\bar{\beta }$ are obtained. The final attention weight values α and β are calculated through SoftMax normalization, and ⊕ represents vector concatenation. Additionally, in the visual attention mechanism, the parameters $q_{att}$, $W_{att}$, and $b_{att}$ are represented as attention parameters shared among different samples, and they are obtained through training and learning.

To enhance the stability of model predictions and reduce output variance, this study employed a multi-head attention mechanism to perform multiple calculations on the fusion between views. Each calculation is conducted independently, and the results of these multiple calculations are finally concatenated. If the view fusion feature map obtained through k calculations of the attention mechanism, with a dimension of d is denoted as $F_{1},F_{2},F_{3},\cdots ,F_{k}\in R^{N\times d}$, then the result of the multi-head attention calculation, denoted as $\bar{F}$, can be represented by Formula (10):(10)$$\begin{array}{c} \bar{F}=F_{1}⊕F_{2}⊕F_{3}\cdots ⊕F_{k}\in R^{N\times kd} \end{array}$$

Here, ⊕ represents the tensor concatenation operation. After introducing the multi-head attention mechanism, the dimension of the view fusion feature map increases to *k* times the original attention mechanism dimension.

### 2.3. Pavement Internal Disease Prediction

In order to better understand the relationship between the feature maps obtained after view fusion and disease categories, as well as to further improve the expressive ability of the model, a multilayer perceptron (MLP) and a SoftMax classification layer are added to the representation layer of the dual-tower model for disease category prediction. This MLP mainly includes two fully connected layers and the activation function ReLU for nonlinear transformation. Given a view fusion feature map $\bar{F}$ with the application of the multi-head attention mechanism, the hidden layer $H_{hid}$ and the model’s disease output prediction value C are represented by Formulas (11) and (12):(11)$$\begin{array}{c} H_{hid}=RELU\left(W_{hid}\bar{F}+b_{hid}\right) \end{array}$$
(12)$$\begin{array}{c} \bar{Y}=SoftMax\left(W_{out}H_{hid}+b_{out}\right) \end{array}$$

The weight values $W_{hid},W_{out}$ and the bias values $b_{hid},b_{out}$ are parameters in the MLP, obtained through model training. To describe the degree of difference between the model’s prediction results and the actual disease categories, this study adopted the cross-entropy loss function. If the true label for the i-th sample in the training set is $Y_{li}$, and the algorithm’s predicted output is ${\bar{Y}}_{li}$, then the cross-entropy loss function is calculated as shown in Formula (13):(13)$$\begin{array}{c} {ℒ}_{\mathrm{cross}-\mathrm{entropy}}=-\underset{l\in L}{\sum }\overset{n}{\underset{i=1}{\sum }}Y_{li}\ln{\bar{Y}}_{li} \end{array}$$

Here, L represents the training set, and n represents the number of sample categories. During the training process, the cross-entropy loss function is used for the backpropagation of gradients to update all learnable parameters in the twin-tower model, including the feature extractor, channel attention mechanism, and inter-view attention mechanism.

## 3. Experimental Results and Discussion

### 3.1. Internal Pavement Disease Dataset

To validate the effectiveness and accuracy of the proposed twin-tower model, extensive experiments were conducted. The road image data used in the experiments originated from the results of 3D GPR surveys conducted on urban roads in Zhejiang Province. Each road sample in the collected raw data contains two image datasets: a B-Scan grayscale image of 320 × 320 pixels and a C-Scan grayscale image of 320 × 230 pixels.

Due to the differences in road disease characteristics between the C-Scan and B-Scan views, Table 2 presents the results of a statistical analysis of the distribution of disease characteristics in the two views of the data. Example images of various disease characteristics are displayed in Figure 5 (from left to right: elongated, dark spots, bright spots, and normal; and hyperbolas, highlights, and normal). Subsequently, based on the disease characteristics presented in the C-Scan and B-Scan images, all collected road radar samples can be classified into four categories using the disease determination flowchart: cavities, interlayer gaps, cracks, and normal roads without diseases. The data statistics for each disease category are shown in Table 3. In the experiment, the overall road radar disease dataset is divided into training, validation, and test sets, at a ratio of 0.5:0.25:0.25, based on the C-Scan view data, which have a larger number of categories.

### 3.2. Experimental Parameters

The experiments in this study were conducted using the PyTorch 2.4 framework, with an RTX 3090 GPU, and the operating environment was Python 3.10. The batch size for all experiments was set to 32, the number of epochs was set to 80, and the Adam optimizer was used for parameter optimization. Only the training set was used for parameter updates; furthermore, the model with the best performance on the validation set was selected, and its average accuracy and standard deviation on the test set were reported. The same experimental setup was repeated 10 times. Due to the large difference in the number of samples in the different categories, the weighted average recall and weighted average F1 score were also calculated to evaluate the model. Given that a positive sample predicted as positive is TP, a negative sample predicted as negative is TN, a negative sample predicted as positive is FP, and a positive sample predicted as negative is FN, the calculation formulas for each evaluation metric in a single experiment are shown in Equations (14)–(17):(14)$$\begin{array}{c} \;Accuracy=\frac{TP+TN}{TP+TN+FP+FN} \end{array}$$
(15)$$\begin{array}{c} \;Precision=\frac{\;TP}{\;TP+FP} \end{array}$$
(16)$$\begin{array}{c} Recall=\frac{\;TP}{\;TP+FP} \end{array}$$
(17)$$\begin{array}{c} F1=\frac{2\times Precision\times Recall}{Precision+Recall} \end{array}$$

Comparative methods were used in the experiments. For Scheme 1, a six-layer simple convolutional neural network (CNN-6), a Residual Network (ResNet18), DenseNet121, and FCN were used as classifiers, and the classification process was carried out using the procedure in Figure 6a. For Scheme 2, the multi-view recognition algorithm model proposed in this study (Dual-AttFusioNet) was used for classification prediction. The main process is shown in Figure 6b, where two classifiers are trained separately, and the final disease prediction category is obtained based on the outputs of the two classifiers.

### 3.3. Original Data Experimental Results

The experimental results of Scheme 1 and Scheme 2 in this study are summarized in Figure 7.

Figure 7 clearly illustrates that, when evaluated on the same test set, the end-to-end prediction approach (Scheme 2), which leverages various fusion techniques, significantly outperforms the step-by-step prediction method (Scheme 1) that involves training separate classifiers for each view. The performance enhancement is attributed to the end-to-end method’s capability to fully harness data from both perspectives. Our proposed multi-view recognition algorithm, Dual-AttFusioNet, which is based on an attention fusion mechanism, achieves the top performance in Scheme 2. It boasts a classification accuracy of 94.62% on the original dataset, an 8.67% improvement over the DenseNet-based step-by-step method. Moreover, Dual-AttFusioNet achieves a recall of 94.79% and an F1 score of 94.64%, outperforming other algorithms in terms of performance. Notably, Dual-AttFusioNet also exhibits high output stability, with the smallest variance in classification accuracy. By incorporating multiple attention fusion mechanisms—such as channel-wise attention during feature extraction and element-wise attention during feature fusion—Dual-AttFusioNet effectively extracts crucial disease feature information from the sample data and performs targeted fusion for each sample, thereby enhancing the overall fusion effect. Consequently, the model proposed in this paper excels in disease classification tasks, confirming the efficacy of our proposed scheme.

### 3.4. Enhanced Data Experimental Results

The experiments mentioned above were conducted on the original radar road data. However, as indicated by the data statistics in Table 2, the overall number of samples in the original dataset is relatively small, which is disadvantageous for training deep learning models. Additionally, the category distribution is highly uneven. For instance, in C-Scan, the number of dark spot datasets labeled as 1 is 129, whereas the number of bright spot datasets labeled as 2 is only 46. The quantities of labels 1 and 2 are much smaller than those of other types of data. This data imbalance causes the model to be biased toward outputting labels with more samples, which affects the accuracy of training.

To address the issues of limited annotated samples and imbalanced disease category distribution in real-world data, as well as to further improve the performance of the model, this study conducted data augmentation and upsampling processing based on the original dataset. As C-Scan has more categories and the disease feature distribution is more imbalanced, data upsampling was first performed according to the category distribution of C-Scan to balance the distribution of each of its categories. Then, the corresponding B-Scan data were also subjected to upsampling. After the upsampling operation, data augmentation was used to expand the dataset. After testing various data augmentation methods, brightness transformation and flipping operations were finally selected to triple the size of the balanced dataset. The final road radar dataset is obtained after upsampling and data augmentation, and the statistical distribution of C-Scan is shown in Table 4.

The model proposed in this paper was retested on the enhanced C-Scan dataset of internal pavement diseases, using the training set for training, and it was also compared with the methods listed in Figure 8. The test results of various methods on the enhanced dataset are summarized in Figure 8.

In Figure 8, it can be seen that, on the enhanced dataset, Scheme 2 still has a significant advantage over Scheme 1. At the same time, the deeper neural network models exhibit a marked improvement in performance when trained on the enhanced dataset, demonstrating enhanced accuracy and stability. This enhancement can be attributed to data augmentation, which partially mitigates the challenges posed by a limited number of training samples and sample imbalance. The Dual-AttFusioNet model introduced in this study continued to outperform its counterparts, achieving even better results on the augmented dataset. It reached an accuracy of 95.78%, a recall rate of 95.87%, and an F1 score of 95.78%. Additionally, the precision increased by 1.16% compared to the results obtained on the original dataset. These findings provide further evidence of the efficacy of our proposed approach.

### 3.5. Experimental Results of Algorithm’s Generalization Ability

To verify the generalization capability of the Dual-AttFusioNet algorithm proposed in this paper, experiments were conducted under highway and urban road conditions. Ground-penetrating radar was used to collect images of internal pavement diseases for stone mastic asphalt (SMA) and asphalt concrete (AC) mixtures. Based on these data, the algorithm presented in this paper was applied for the intelligent identification of internal asphalt pavement diseases. The results of the algorithm’s identification are shown in Figure 9 and Figure 10.

In Figure 9 and Figure 10, it can be seen that the environment in which the highway is located has a significant impact on the algorithm’s recognition effect. On highways, the average accuracy, recall, and F1 score of the algorithm for identifying internal pavement diseases is 91.78%, 93.26%, and 92.68%, respectively. However, on urban roads, the algorithm’s performance in recognizing internal pavement diseases is relatively poor, with average accuracy, recall, and F1 scores of 84.03%, 86.18%, and 85.46%. This is due to the numerous underground pipelines and rainwater drains on urban roads, which cause interference with ground-penetrating radar imaging, leading to a decrease in the accuracy of the algorithm’s identification of internal pavement diseases.

### 3.6. Experimental Results of Multi-view Fusion

To verify the effect of multi-view fusion on improving model performance, this study conducted separate experiments with B-Scan and C-Scan, as well as with their combination. The performance of the model was evaluated through accuracy, recall rate, and F1 score, as shown in Figure 11.

In Figure 11, it can be seen that, when using a single B-Scan view, the algorithm achieved accuracy, recall, and F1 scores of 85.35%, 85.04%, and 82.2%, respectively. When employing a single C-Scan view, the accuracy, recall, and F1 scores were 82.11%, 81.87%, and 81.79%, respectively. However, after combining both B-Scan and C-Scan views, the algorithm’s performance significantly improved, reaching accuracy, recall, and F1 scores of 94.62%, 94.79%, and 94.64%. These experimental results fully demonstrate the superiority of multi-view integration, which overcomes the defects of single-view approaches in missing key image features and enhances the algorithm’s accuracy through complementary feature information.

## 4. Conclusions

In this paper, we propose an end-to-end deep attention twin-tower model that utilizes both views of a multi-view radar spectrum dataset collected using 3D ground-penetrating radar, targeting the road disease category prediction task. Firstly, two feature extractors with different parameters were trained for the two types of view information, which can enhance the network’s ability to represent a single view. Moreover, a channel-wise attention mechanism was introduced during feature extraction to learn the weight relationships between channels, further improving the network’s feature representation capability. Finally, an attention mechanism was used again to learn the weight relationships element by element when fusing the two views, enhancing the weight values of the important views and improving prediction accuracy.

Experiments showed that the algorithm proposed in this paper achieves superior prediction results on the radar spectrum dataset, with a prediction accuracy that is approximately 8% higher than that of the distribution prediction based on the disease identification flowchart. The double application of channel-wise and element-wise attention fusion mechanisms can effectively improve the model’s ability to represent road disease information, reduce the loss of feature information, and improve the accuracy of image information processing. The algorithm proposed in this paper is sensitive to environmental influences, achieving an accuracy rate of 91% for the identification of internal diseases on highways but only 84% on urban roads. Therefore, in future work, enhancing the algorithm’s generalization ability and resistance to interference will be a key focus.

## Figures and Tables

**Figure 1 sensors-24-06757-f001:**
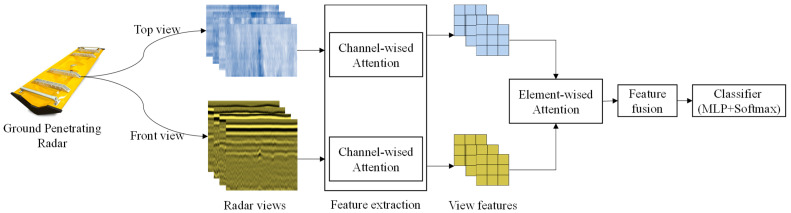
Multi-view recognition framework.

**Figure 2 sensors-24-06757-f002:**
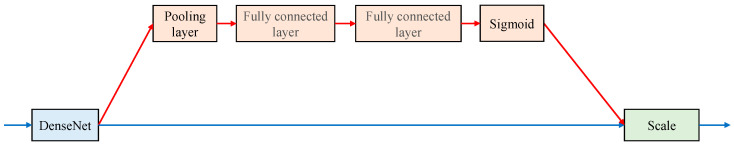
Inception-based SE structure.

**Figure 3 sensors-24-06757-f003:**
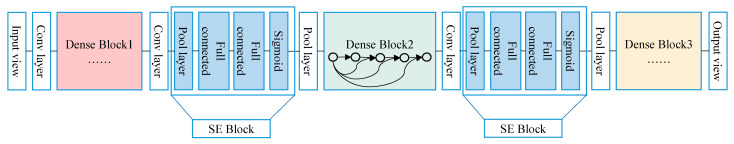
SE-DenseNet blocks.

**Figure 4 sensors-24-06757-f004:**
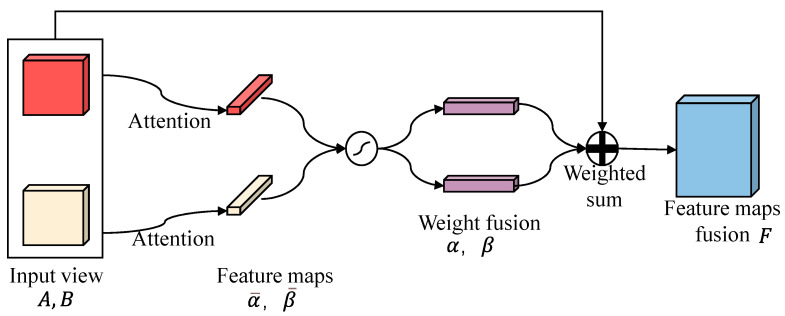
Element-wised attention feature fusion.

**Figure 5 sensors-24-06757-f005:**
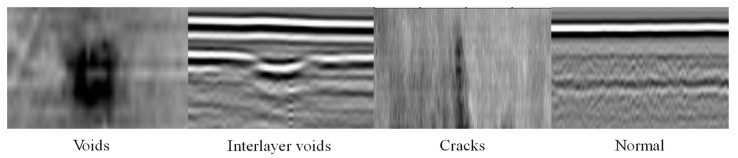
Example images of internal road diseases.

**Figure 6 sensors-24-06757-f006:**
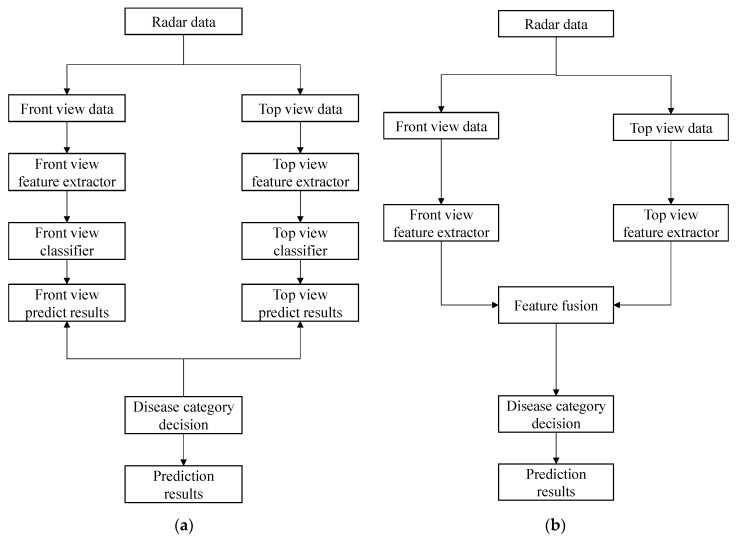
Algorithm flowchart. (**a**) Traditional algorithm. (**b**) Algorithm proposed in this study.

**Figure 7 sensors-24-06757-f007:**
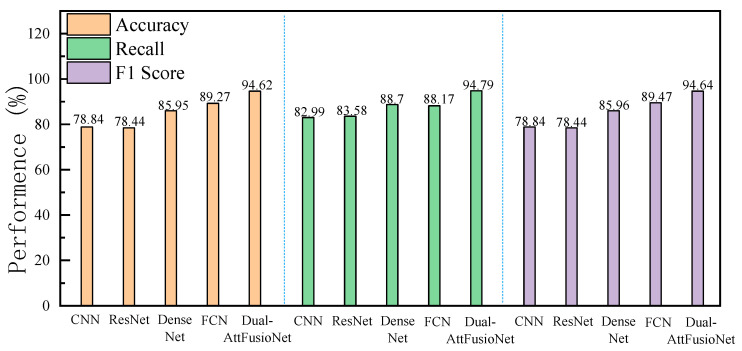
Original data experimental results.

**Figure 8 sensors-24-06757-f008:**
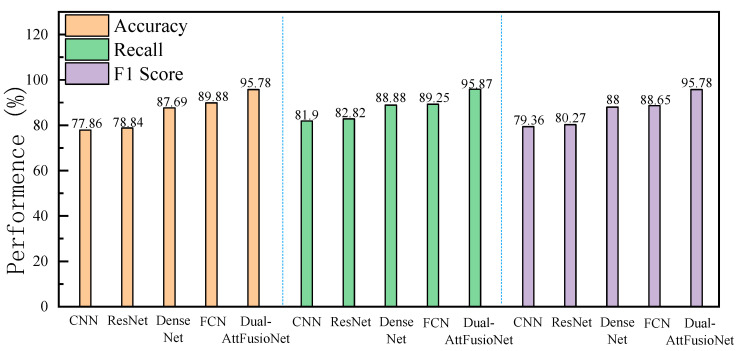
Enhanced data experimental results.

**Figure 9 sensors-24-06757-f009:**
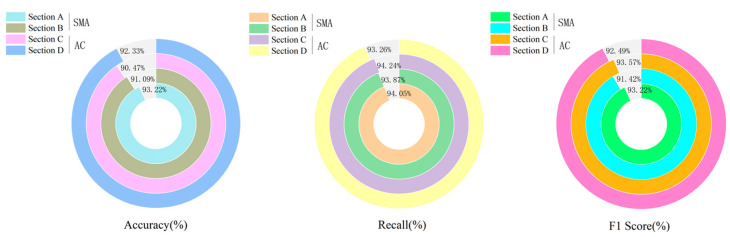
Identification results of internal pavement diseases on highways.

**Figure 10 sensors-24-06757-f010:**
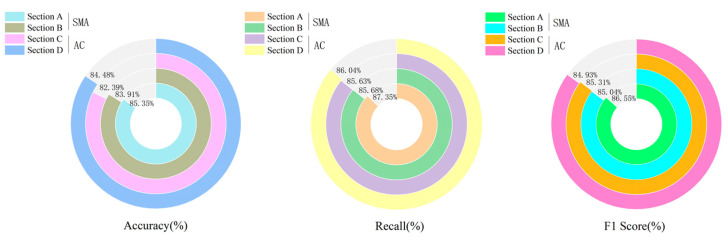
Identification results of internal pavement diseases on urban roads.

**Figure 11 sensors-24-06757-f011:**
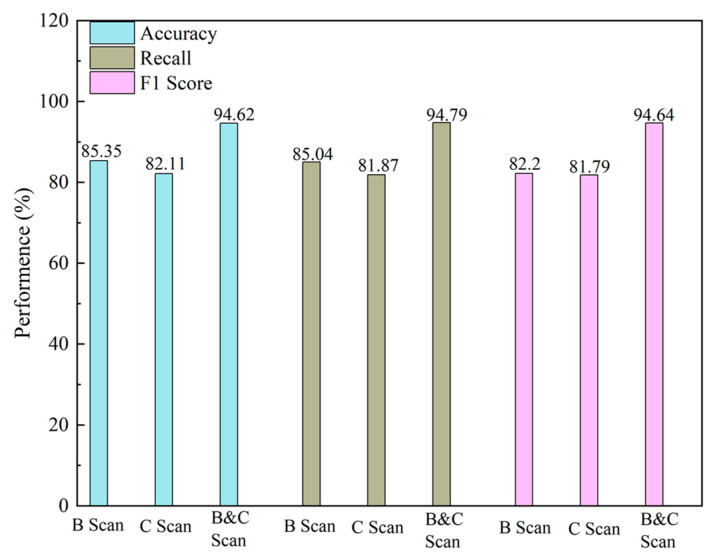
Comparison results of single-view and multi-view approaches.

**Table 1 sensors-24-06757-t001:** Network structure of DenseNet-121.

Layers	Image Size	Structure		
Convolution	112 × 112	7 × 7 conv, stride 2		
Pooling	56 × 56	3 × 3 Max pool, stride 2		
Dense Block	56 × 56	[	1 × 1 conv	] × 6
			3 × 3 conv	
Transition Layer	56 × 56	1 × 1 conv		
	56 × 56	SE module		
	28 × 28	2 × 2 Avg. pool, stride 2		
Dense Block	28 × 28	[	1 × 1 conv	] × 12
			3 × 3 conv	
Transition Layer	28 × 28	1 × 1 conv		
	28 × 28	SE module		
	14 × 14	2 × 2 Avg. pool, stride 2		
Dense Block	14 × 14	[	1 × 1 conv	] × 24
			3 × 3 conv	
Transition Layer	14 × 14	1 × 1 conv		
	14 × 14	SE module		
	7 × 7	2 × 2 Avg. pool, stride 2		
Dense Block	7 × 7	[	1 × 1 conv	] × 16
			3 × 3 conv	
Representation Layer	7 × 7	SE module		
	1 × 1	7 × 7 global Avg. pool		
		Fusion, Fully connected, SoftMax		

**Table 2 sensors-24-06757-t002:** Radar views of internal pavement diseases.

Label	Feature	Quantity
0 C-Scan	Long strip	307
1 C-Scan	Dark spot	129
2 C-Scan	Bright spot	46
3 C-Scan	Normal	200
0 B-Scan	Hyperbola	309
1 B-Scan	Highlight	173
2 B-Scan	Normal	200

**Table 3 sensors-24-06757-t003:** Internal pavement diseases.

Label	Label Name	Quantity
0	Voids	173
1	Interlayer voids	23
2	Cracks	286
3	Normal	200

**Table 4 sensors-24-06757-t004:** Enhanced C-Scan dataset of internal pavement diseases.

Label	Label Names	Law Data Quantity	Training Data Quantity after Upsampling	Training Data Quantity after Augmentation
0	Voids	86	143	562
1	Interlayer voids	11	143	562
2	Cracks	143	143	562
3	Normal	100	143	562

## Data Availability

Data are contained within the article.

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
