# Peer review of "Research on Road Internal Disease Identification Algorithm Based on Attention Fusion Mechanisms"

_sensors, 2024, doi:10.3390/s24206757_

Round 1
Reviewer 1 Report
Comments and Suggestions for Authors
The manuscript aims to enhance the accuracy and speed of identifying internal pavement diseases by developing a deep learning-driven, multi-view recognition algorithm model. This model integrates attention mechanisms across different channels and perspectives to improve the capture and analysis of pavement disease characteristics. However, there are several issues with the content of the research work:
- The manuscript contains a number of writing errors, and the author is urged to check the full text carefully, e.g., “h natural” in line 28, “..” in line 23.
- The manuscript lacks a visual analysis of the detection results, which is insufficient to ensure the completeness of the paper.
- The manuscript is missing a comparison with state-of-the-art (SOTA) methods; CNN, ResNet, FCN, and Dual-attention are relatively outdated approaches.
- The introduction section of the manuscript lacks a motivational analysis for using attention mechanisms and fails to explain why they are employed.
- The research subject is the detection of internal diseases in asphalt pavements, but the author has not analyzed the characteristics of these internal diseases, nor have they discussed the benefits that attention mechanisms could bring.
- In the Methodology section, there is an extensive introduction to existing knowledge points such as DenseNet and attention mechanisms, but it fails to clarify the author's innovative contributions within the framework of Figure 1.
- The images in the manuscript are almost all in black and white; the author is requested to enhance the visual appeal of the network diagrams.
- There are some issues with the formulas in the manuscript, such as the inconsistency in the expression of L in Equation (13), what is the relationship between Yl and Yli ?
In summary, the author's understanding of internal diseases in asphalt pavements appears to be insufficient, leading to a poorly formulated research question. The methods used lack innovation, and the comparative experimental analysis is inadequate, which undermines the completeness of the research content.
Comments on the Quality of English LanguageExtensive Englishe editing required.
Reviewer 2 Report
Comments and Suggestions for Authors
Road Internal Disease should be replaced by Road Health Monitoring or Structural defect monitoring in Roads/ Pavements. This will attract civil engineers to this important article.
Idea is application of algorithms for Road is partly novel.
end-to-end deep attention twin-tower model that utilizes both views of the multi-view radar spectrum dataset collected by 3D ground-penetrating radar, targeting the road disease category prediction task, is interesting.
Comments on the Quality of English Languageplease correct some minor english errors/typos
Reviewer 3 Report
Comments and Suggestions for Authors
The paper is about using fusion modules of deep learning feature extraction networks to classify the internal pavement diseases. The tile is not tightly represented this main idea, and there are major problems in this research:
1. Technical problems:
1) Those citations with authors’ names should be added with “et al.”. Those abbreviations appear first time should be explained, including YOLO, GPR, R-CNN, IOU, etc.
2) What is the contribution of the references cited in the paper to the proposed method? If not, why need them? If so, please state which aspects are helpful to develop this research in the paper?
3) The portion for DenseNet-121 is not clear to me, and I suppose this is not the focus of this paper, but authors can address it in such a detail if it is developed by yourselves, otherwise, readers can investigate it by themselves if interested and the authors should give the reference source.
4) Readers are not experts in this field, the background information for internal pavement diseases and how to identify them with different-view scanning. Therefore, it is more friendly if the authors can show some defective pavements within the available data to readers, since visual data are used here.
5) Figure 2 can be rotated for better view, but the meaning of arrows should be pointed out. I do not get any point form Figure 3 to explain SE-DenseNet, although there are a lot of text on it. In figure 4, what are the various cubes, arrows, and symbols standing for? In Figure 5, What is each subfigure? B-Scan or C-Scan?
6) How did authors collect and annotate the dataset used in the paper? Why they are credential to this research? Each internal distresses should be given examples.
7) What if no fusion is used for the proposed method? Or what are the results if we rely on one of the views (B- or C-Scan) for disease recognition? Furthermore, what if the other weights of different-view features fused in the framework are used? How about the cost (data collection and processing) and efficiency of the proposed work? What is the challenges, although its performance is as high as 94.6%?
2. Language problems:
1) This paper is not readable at all because of the poor writing, including the organization, grammar mistakes, and wordy. For examples, the first sentence of the introduction is not a complete one. “three-dimensional” should be 3D.
2) In Table 1, what are meanings of layers, size, and structure? Can these be indicated on the Figure 1 or somewhere?
3) Each symbol in equations should be explained when they first show up. In addition, some of vectors or tensors should be bold in text. And keep their definitions consistent in case of causing confusion to readers.
4) In equation, the dot or cross product between these vectors or variables should be with “.” or “x”. Please pay attention to the scientific formats of them.
Comments on the Quality of English Language2. Language problems:
1) This paper is not readable at all because of the poor writing, including the organization, grammar mistakes, and wordy. For examples, the first sentence of the introduction is not a complete one. “three-dimensional” should be 3D.
2) In Table 1, what are meanings of layers, size, and structure? Can these be indicated on the Figure 1 or somewhere?
3) Each symbol in equations should be explained when they first show up. In addition, some of vectors or tensors should be bold in text. And keep their definitions consistent in case of causing confusion to readers.
4) In equation, the dot or cross product between these vectors or variables should be with “.” or “x”. Please pay attention to the scientific format for them.
Round 2
Reviewer 1 Report
Comments and Suggestions for Authors
This study developed a deep learning-driven multi-view recognition algorithm model, which enhances the ability to capture and analyze pavement defect features by integrating attention mechanisms across different channels and perspectives. I am pleased to see that the authors have made revisions to the manuscript based on the feedback; however, the following issues still need to be addressed:
- The authors have provided an explanation for the parameters in equation (13), but further clarification of the notation used for different symbols needs to be added to the manuscript.
- There are still some grammatical errors in the manuscript that should be carefully addressed. For example, "YOLOV2" and "YOLOv4" have inconsistent capitalization of the letter "V". Additionally, there are inconsistencies between italic and non-italic variable formatting in the equations, and Table 1 is incorrectly formatted.
- The introduction lacks a thorough analysis of why the attention mechanism was used. I have reviewed the response provided by the authors, but it does not sufficiently justify how the addition of an attention mechanism improves defect detection. Please supplement the manuscript with relevant research on attention mechanisms.
- The references in the manuscript are relatively outdated, with a low percentage of articles from the past five years. Please reorganize the introduction taking into account the suggestions in comment 3.
- In Figure 4, what is the difference between the two Attention mechanisms? Is it spatial attention, channel attention, or something else? Please ensure that the figure representation is clear and accurate.
- The manuscript lacks visualization of test results and comparative analysis.
Extensive editing of English language required.
Reviewer 3 Report
Comments and Suggestions for Authors
Thanks for the improvement on English writing and quality of presentation.
Comments on the Quality of English LanguageNo comments.
